# Simultaneous Monitoring of Multi-Enzyme Activity and Concentration in Tumor Using a Triply Labeled Fluorescent In Vivo Imaging Probe

**DOI:** 10.3390/ijms21093068

**Published:** 2020-04-27

**Authors:** Jenny Tam, Alexander Pilozzi, Umar Mahmood, Xudong Huang

**Affiliations:** 1Wyss Institute and Harvard Medical School, Boston, MA 02115, USA; jenny.tam@wyss.harvard.edu; 2Neurochemistry Laboratory, Department of Psychiatry, Massachusetts General Hospital and Harvard Medical School, Charlestown, MA 02129, USA; apilozzi@mgh.harvard.edu; 3Department of Radiology, Massachusetts General Hospital and Harvard Medical School, Charlestown, MA 02129, USA; umahmood@mgh.harvard.edu

**Keywords:** cathepsin B, matrix metalloprotease-2, biomarker, near-infrared fluorescent probe, molecular imaging

## Abstract

The use of fluorescent imaging probes that monitor the activity of proteases that experience an increase in expression and activity in tumors is well established. These probes can be conjugated to nanoparticles of iron oxide, creating a multimodal probe serving as both a magnetic resonance imaging (MRI) agent and an indicator of local protease activity. Previous works describe probes for cathepsin D (CatD) and metalloproteinase-2 (MMP2) protease activity grafted to cross-linked iron oxide nanoparticles (CLIO). Herein, we have synthesized a triply labeled fluorescent iron oxide nanoparticle molecular imaging (MI) probe, including an AF750 substrate concentration reporter along with probes for cathepsin B (CatB) sand MMP2 protease activity. The reporter provides a baseline signal from which to compare the activity of the two proteases. The activity of the MI probe was verified through incubation with the proteases and tested in vitro using the human HT29 tumor cell line and in vivo using female nude mice injected with HT29 cells. We found the MI probe had the appropriate specificity to the activity of their respective proteases, and the reporter dye did not activate when incubated in the presence of only MMP2 and CatB. Probe fluorescent activity was confirmed in vitro, and reporter signal activation was also noted. The fluorescent activity was also visible in vivo, with injected HT29 cells exhibiting fluorescence, distinguishing them from the rest of the animal. The reporter signal was also observable in vivo, which allowed the signal intensities of the protease probes to be corrected; this is a unique feature of this MI probe design.

## 1. Introduction

Genomic and proteomic approaches have identified a host of molecular markers associated with disease [1,2,3,4]. A central challenge in contemporary biomedical research is the characterization of these factors in the context of the entire organism. Molecular imaging (MI) techniques hold great promise for mapping molecular activities in living animals, but previously reported probes are thus greatly limited in their ability to measure multiple activities simultaneously. Herein, we report the preparation of a fluorescence-based, in vivo optical imaging probe bearing three fluorescent reporters, two of which are responsive to specific protease activities.

Fluorescence-based imaging probes have been fabricated previously using a high molecular weight graft polymer on which fluorochromes were conjugated to the polymer backbone. The fluorescence from these probes was initially quenched until a particular protease cleaved the polymer backbone. Prior publications report on such probes to monitor CatD protease activity [5], MMP2 [6], and thrombin [7]. Another type imaging probe that has been fabricated previously uses iron oxide nanoparticles as a combined optical imaging and magnetic resonance (MR) agent and, in doing so, becomes multimodal [8,9,10].

A dual-fluorochrome imaging probe using iron oxide nanoparticles was described previously [11], with both enzymatic activity through a fluorescently-labeled cleavable enzyme substrate and, in vivo, via a substrate concentration through a non-cleavable internal standard. The use of these probes initially yielded fluorescence as a function of the intensity of the light used, its depth and the site of interest, and the enzyme activity and delivery of the probe (local substrate concentration) [11,12]. Here, we report on an improvement and extension of our previous dual fluorochrome by creating a triple fluorochrome probe (TFP), containing one fluorochrome to report on the local substrate concentration and two fluorophores to monitor the local activity of two enzymes, CatB and MMP2. Unlike previous synthetic strategies employed to create similar imaging probes, the technique outlined in this report pre-labels the peptide substrates prior to the conjugation of the nanoparticle scaffold. The peptide substrates are then conjugated to the nanoparticle surface, while the reporter fluorochrome (for probe concentration) is attached to the nanoparticle through a proteolytic-resistant linkage. The ratio of fluorescence due to the enzymatic cleavage of each substrate to the fluorescence of the reporter fluorochrome reflects activation by that particular protease and could be used to correct for differences in the size and depth of the target lesions. By using this method, we are able to, simultaneously in vivo, image multiple enzyme activities and multiple molecular parameters.

## 2. Results

Prior to the synthesis of the TFP probe, the specificity of the peptide substrates (C-peptide, M-peptide) was determined with CatB and MMP2 enzymes (Figure 1). The molecular weights and cleavage sites of these two peptide substrates are in Appendix A.

As expected, MMP2 recognized the M-peptide substrate and did not cleave the C-peptide substrate. Compared to the elution time of the original M-peptide substrate, the high-performance liquid chromatography (HPLC) chromatogram indicates cleavage of the M-peptide substrate by MMP2. Since the M-peptide substrate is not a substrate for CatB, the peptide remained intact and eluted at the original time when incubated with CatB. When incubating the MMP2 enzyme with the C-peptide substrate, the C-peptide substrate had the same elution time as the control and the uncleaved C-peptide substrate. However, the CatB enzyme recognized the C-peptide substrate, removing the original C-peptide peak from the HPLC chromatogram.

The structure of the TFP, as shown in Figure 2, illustrates the iron oxide nanoparticle scaffold conjugated to the fluorescently-labeled M-peptide and C-peptide substrates as well as the reporter dye that reports on the delivery of the TFP probe (AF750).

The absorption spectrum of the TFP imaging probe is shown in Figure 3 and has an absorption peak at 755 nm (corresponding to the reporter dye), 675 nm (corresponding to the Cy5.5 labeled C-peptide), and 555 nm (corresponding to the AF546 labeled M-peptide). The mass spectra of AF546 labeled M-peptide and Cy5.5 labeled C-peptide are shown in Appendix A, respectively. The spectrum of the amino-CLIO is also shown in Figure 3. Iron oxide, represented as a dotted line, dominates the spectrum in the ultraviolet region, and whose little absorption is in/near the infrared region of the spectrum.

Using the extinction coefficients for the dyes and iron, the number of labeled peptide substrates and reporter dyes was calculated to be 9–10 labeled M-peptide substrates, 7–8 labeled C-peptide substrates, and 2–3 reporter dyes per nanoparticle.

Figure 4 shows the effect of treating the TFP imaging probe with pure CatB and MMP2 enzymes. The activation of the fluorescence signal of the TFP imaging probe was tested under optimal conditions for CatB cleavage by using 20 mM of sodium acetate buffer (pH 5). Fluorescence activation was seen almost immediately with the addition of the CatB enzyme (Figure 4A, blue line) of the C-peptide substrate, while minimal fluorescence activation of the M-peptide substrate (Figure 4A, red line) occurred and essentially no fluorescence activation occurred in the reporter dye (Figure 4A, yellow line). The fluorescence activation of the TFP probe was also tested under optimal conditions for MMP2 cleavage by using 25 mM (4-(2-hydroxyethyl)-1-piperazineethanesulfonic acid) (HEPES) (pH 7). After adding the MMP2 enzyme, the fluorescence activation of the M-peptide substrate (Figure 4B, red line) was seen almost instantaneously compared to low to no fluorescence activation of the C-peptide substrate (Figure 4C, blue line) and the reporter dye (Figure 4B, yellow line). Figure 4C summarizes and compares the two experiments and illustrates the selective activation of the enzyme substrates by their respective enzymes with minimal activation of the other substrate.

The fluorescence activation fold was calculated by taking the fluorescence intensity at the end point and dividing it by the fluorescence intensity at its initial point. As the MMP2 peptide substrate was cleaved by the MMP2 enzyme, a fluorescence increase of 13.15 ± 1.22 occurred. Similarly, when the CatB enzyme cleaved the CatB peptide substrate, a fluorescence increase of 12.11± 0.92 was measured. Both experiments show little to no activation of the reporter dye, 1.31 ± 0.25, indicating that the dye remains attached to the iron oxide nanoparticle scaffold. The cross-reactivity of the peptide substrates to the other enzymes was also measured, as shown in Figure 4. The fluorescence activation of the MMP2 peptide substrate by the CatB enzyme was measured to be 2.16 ± 0.12, and the fluorescence activation of the CatB peptide substrate by the MMP2 enzyme was measured to be 2.13 ± 0.14.

The imaging ability of the TFP probe was also evaluated in cell culture. Human HT29 cells were utilized, and a fluorescence microplate reader monitored the activation of the TFP imaging probe. Immunoblotting was done on this cell line to verify the presence of enzymes in this cell line (Figure 5A). Figure 5B shows the activation of both C-peptide and M-peptide substrates in cell culture due to the presence of both CatB and MMP2 enzymes in cell lines.

Although the fluorescence activation of both peptide substrates occurred, no fluorescence activation of the reporter dye occurred. As shown in Figure 5C, the CatB substrate, MMP2 substrate, and reporter dye (AF750) showed an average activation fold 4.80 ± 0.93, 4.29 ± 0.01, and 1.22 ± 0.06, respectively.

Fluorescence from individual cells incubated with the TFP imaging probe was also measured using flow cytometry and compared to non-incubated cells, as shown in Figure 5D. Unlabeled HT29 human cell lines were washed, and the fluorescence was measured using flow cytometery at the Cy5.5 and AF546 wavelengths to correspond to the fluorescence activation of the CatB and MMP2 peptide substrates, respectively. These signals were compared to HT29 cells that were incubated with the TFP probe for 24 hrs. The cells were washed, and fluorescence intensity was measured again by flow cytometery. Fluorescence signals in both Cy5.5 (CatB substrate fluorescence) and AF546 (MMP2 substrate fluorescence) indicated the internalization of the TFP probe and CatB and MMP2 activity within the cells. By comparison, no fluorescence signals were seen in the control cells, indicating that this cell line internalizes the TFP probe.

To demonstrate the ability of the TFP probe to act as an in vivo optical imaging probe, the imaging agent was injected into female nude mice by IV tail vein injection The fluorescence signal was measured and analyzed with respect to CatB and MMP2 activities, namely the Cy5.5 and AF546 signals proportional to the signal of probe concentration (AF750 signal). Uptake of the TFP probe can be seen in Figure 6.

By taking the ratio of the fluorescence activation signal and dividing it by the delivery signal, the corrected activation signal of enzyme activity can be obtained:

(1)
Act_corr_ = Act_raw_ / Delivery


In Figure 6A, the MMP2 fluorescence activation signal and TFP probe concentration fluorescence are seen in the tumor area. No fluorescence signal is seen in the surrounding tissue. By correcting the MMP2 activation signal with the probe concentration signal, MMP2 fluorescence activation is displayed in the tumor independent of delivery concentration. Similarly, in Figure 6B, the CatB fluorescence activation signal and AF750 substrate concentration signal are seen in the tumor area compared to no fluorescence activity seen in the surrounding tissue. Similar to the MMP2 activation signal, the CatB activation signal is corrected by being divided by the probe’s delivery signal (AF750).

## 3. Discussion

Optical imaging probes known as TFPs (triple fluorochrome probes) that can monitor two distinct enzymatic activities and substrate concentration have been designed and synthesized. The wavelengths of the three fluorophores used in the TFP probe are optically distinct with minimal spectral overlap between wavelength regions. In addition, the fluorophores emit in the far red and near-infrared region of the optical spectrum, allowing for greater tissue penetration, while remaining optically distinct for in vivo imaging.

Although these imaging probes were not tested as such, the iron oxide core of the TFP probes has been shown as an MR contrast agent as well as optical probes [13,14]. Previous reports have shown that the subcutaneous administration of iron oxide nanoparticles results in contrast-enhanced MR images of lymph nodes as well as an optical signal that could be detected using the same imaging equipment described in this paper [15]. The MR capability of the TFP probe can be utilized in future applications to provide additional anatomical context to in vivo visualization of probe activation.

Previous fluorescently activatable imaging probes have been synthesized whereby fluorescently labeled peptide substrates were labeled and conjugated to graft copolymers [5,16]. The multiple fluorochromes attached to this graft copolymer are quenched due to the interactions between the dye molecules [5,16,17]. Due to different types of substrates attached to the scaffold, the TFP probe could not use the same graft copolymer platform as a scaffold for the TFP probe. For example, by having two or more different types of enzymatic substrates conjugated to the graft copolymer, a proteolytic cleavage of any of the peptide substrates would cause a fluorescence increase in all of the substrates (cleaved or intact).

Thus, the TFP imaging probe was built upon an iron oxide nanoparticle scaffold surrounded by a CLIO. This nanoparticle scaffold was chosen due to its ability to quench the fluorescence of organic dyes. In the case of the TFP, the iron oxide core quenches the fluorescently labeled peptide substrates until it is cleaved by the substrate’s corresponding enzyme. A fluorescence increase occurs as the labeled, cleaved portion of the substrate leaves the local environment of the nanoparticle and distributes through the bulk media. The fluorescence quenching of the magnetic nanoparticles may be due to the nonradiative energy transfer between the dye and the iron oxide or to collisions between the fluorescence dyes and the nanoparticle [18,19].

The probe design and chemistry offer a flexible design for optically activatable nanoparticles that can include different substrates for other enzymes and multiple enzymatic targets. Unlike previous synthetic strategies employed to create similar imaging probes, the technique outlined in this paper pre-labeled the peptide substrates prior to conjugation to the nanoparticle scaffold. By pre-labeling the peptide substrates with a fluorochrome, multiple enzymatic substrates with distinct optical labels can be conjugated to the iron oxide scaffold. However, as more fluorescent labels are added to the imaging probe, more sophisticated techniques such as fluorescence molecular tomography (FMT) [20,21,22] or spectral unmixing techniques [23,24,25] can increase fluorescence sensitivity or further refine and distinguish between similar optical channels, respectively.

The affixing of polyarginyl-containing regions to peptide substrates that are attached to nanoparticles has increased translocation through cell membranes. Internalization can be accomplished through the use of positively charged peptide signals, such as those derived from human immunodeficiency virus (HIV) trans-activator of transcription (Tat) protein, or homeoprotein transcription factor [26,27,28]. CLIO nanoparticles have been conjugated to portions of the tat peptide sequence, and these Tat-CLIO nanoparticles have translocated within cells [26,29,30]. Membrane translocating activity appears to be primarily dependent on the headgroup of arginine [31,32,33], so nanoparticles conjugated to peptides with simply polyarginyl regions may efficiently enter cells, as was seen with flow cytometry data (Figure 5C).

Using a single particle, the TFP multimodal imaging probe would be able to gather through in vivo imaging, in addition to lesion size and depth, new and more types of information simultaneously. By adding an optical channel that monitors the delivery of the probe, the probe can indicate its transport and concentration within the vicinity of the target as the other two fluorochromes monitor the activity of the probe interacting with its molecular target, which, in this case, is a protease. Various physiological factors, including blood flow as well as capillary permeability and volume, can affect probe transport [34,35]. These studies have shown that the TFP activation can provide information independent of the absolute fluorescence of the other two optical channels. Thus, the reporter fluorochrome (AF750) provides an internal standard for determining probe concentration and allows fluorescence from protease activity (Cy5.5 for CatB and AF546 for MMP2) to be corrected for variable levels of probe transport.

The elevated levels of enzymatic activity of CatB and MMP2 are linked to a variety of medical conditions, such as cancer metastasis [36,37,38,39]. The development of enzymatic diagnostic nanoparticles may be realized due to similar particles that are clinically used and accumulated in the liver, spleen, and lymph node macrophages. Designing the probe described in this study may provide a more accurate and global picture of enzymatic activity related to certain diseases given its ability to obtain satisfactory optical images of multi-enzymatic activity in vivo. Future imaging probes of this type might be developed for clinical use.

## 4. Materials and Methods

### 4.1. Preparation of a TFP Imaging Probe

An imaging probe containing two types of fluorescently activatable peptide substrates and one type of reporter fluorochrome was synthesized according to Figure 1 and denoted as the Triple Fluorochrome Probe (TFP). “Triple” denotes that the imaging probe can monitor three fluorescence channels, while CLIO nanoparticles are used as a carrier scaffold. The primary amine of a dextran-coated CLIO (amino-CLIO) was first fluorescently labeled with a fluorochrome (reporter dye), creating “fluoro-CLIO” that monitors the delivery of the nanoparticle in vivo, and is then subsequently conjugated with fluorescently-labeled peptide substrates. The peptide substrates were conjugated to the nanoparticle using an activating agent, succinimidyl iodoacetate (SIA), which links the peptide substrates through the cysteine residue to the nanoparticle surface (Figure 1).

#### 4.1.1. Preparation of Fluorescently Labeled CLIO Nanoparticle

Obtained from Lee Josephson of Massachusetts General Hospital (MGH), Boston, MA, USA, [29,40], aminated CLIO consists of a core of superparamagnetic iron oxide coated with a layer of cross-linked dextran that has a layer of primary amines. The CLIO nanoparticles were fluorescently labeled with the reporter dye that is a constitutively active fluorophore that monitors the delivery of the nanoparticle in an in vivo animal model (fluoro-CLIO). To label the CLIO nanoparticles, 1 mg of Alexa Fluor 750 (AF750) (Invitrogen Co., Carlsbad, CA, USA) was dissolved in dimethylsulfoxide (DMSO) (Sigma-Aldrich, St. Louis, MO, USA) and was mixed with 1 mg of amino-CLIO and 108 mmol triethylamine (TEA) (Sigma-Aldrich) at room temperature overnight in the dark to create the dye-labeled fluoro-CLIO. The fluoro-CLIO was then purified using Sephadex G-25 columns (GE Healthcare, Piscataway, NJ, USA) and equilibrated with 1X phosphate buffered saline (PBS) (Fisher Scientific, Hampton, NH, USA).

#### 4.1.2. Synthesis of Peptide Sequences as Enzymatic Substrates

The following peptide sequences were synthesized by the Tufts University Peptide Core Facility (Boston, MA):

for cathepsin B (C-peptide): NH_2_–G–G–R–R–G–G–C–COOH

for MMP2 (M-peptide): NH_2_–G–V–P–L–S–L–S–G–r–r–r–C–COOH

The “r–r–r” sequence denotes d-arginines, while the other amino acid residues are L-residues.

#### 4.1.3. Fluorescently Labeling Peptide Sequences

The peptide substrates were each labeled with a distinct fluorochrome: the cathepsin B (CatB) peptide substrate (C-peptide) was labeled with Cy5.5 dye (GE Healthcare), and the MMP2 peptide (M-peptide) was labeled with Alexa Fluor 546 (AF546) (Invitrogen Co). Each peptide substrate was mixed in its own reaction vessel, and each vessel contained the peptide substrate dissolved in DMSO (5 mg/mL), the fluorophore dissolved in DMSO (5 mg/mL), and 108 mmol of TEA. The reaction was mixed and allowed to sit at room temperature in the dark overnight. The peptides’ substrates were then purified through reverse-phase high-performance liquid chromatography (HPLC) apparatus (ProStar, Varian Inc., Palo Alto, CA) using a preparative scale column (C18, 10 mm, 250 mm × 21.2 mm, Grace-Vydac, Deerfield, IL, USA) at a flow rate of 7 mL/min.

#### 4.1.4. Conjugation of Labeled Peptide Sequences to a CLIO Nanoparticle

To conjugate the fluorescently-labeled peptide sequences to the nanoparticle surface, the fluoro-CLIO nanoparticle surface was first activated by adding 0.5 mL of 150 mM of a linker, succinimidyl iodoacetate (SIA) (Molecular Biosciences, Boulder, CO, USA), to fluoro-CLIO, thereby forming fluoro-iodoacetyl-CLIO. The reaction sat for 30 min at room temperature. Fluoro-iodoacetyl-CLIO was separated from the excess iodoacetic acid using a Sephadex G-25 column equilibrated with 1X PBS.

After purification, the 1 mg of the fluorescently labeled M-peptide substrate was mixed with 1 mg of fluoro-iodoacetyl-CLIO for 30 min at room temperature in the dark. Then, 1 mg of the fluorescently labeled C-peptide substrate was added. The entire mixture was allowed to sit at room temperature in the dark overnight. The TFP imaging probe was separated from the excess peptide substrates using a Sephadex G-25 column equilibrated with 10X PBS.

### 4.2. Peptide Specificity Cleavage Assays through HPLC

The enzyme specificity of the C-peptide and MMP2 peptide substrates were tested by incubating each peptide substrate separately with CatB and MMP2 enzymes. A vial containing 1 mg of C-peptide or M-peptide was dissolved in 20 mM sodium acetate buffer, 2 mM dithiothreitol (DTT), and 5 mM of ethylenediaminetetraacetic acid (EDTA) and incubated with 10 mU of CatB enzyme (CatB Bovine, Calbiochem, Gibbstown, NJ, USA) at room temperature for 35 min. Another vial containing 1 mg of C-peptide or M-peptide was dissolved in 25 mM HEPES buffer (pH 7) containing 10 mM calcium chloride and incubated with 0.5 mg MMP2 (MMP2 Human, Calbiochem) at 37 °C for 24 h. A control experiment for C-peptide was conducted whereby 1 mg of C-peptide was dissolved in 20 nM sodium acetate buffer (pH 5), with 2 mM DTT, and 5 mM EDTA for 24 h at 37 °C (no enzyme was added), and 1 mg of M-peptide was dissolved in 25 mM HEPES buffer containing 10 mM calcium chloride and was incubated for 24 h at 37 °C (no enzyme was added). The cleavage of peptide substrates was monitored by reverse-phase HPLC using an analytical scale column (C18, 10 mm, 250 mm × 4.6 mm, Grace-Vydac, Deerfield, IL, USA) at a flow rate of 1 mL/min.

### 4.3. Physical Properties of the Nanoparticles

Using a UV-VIS spectrophotometer (Cary 50 Bio UV-VIS Spectrophotometer, Varian Inc., Palo Alto, CA, USA), the number of fluorescently labeled peptides and constitutively active monitoring fluorophores per CLIO was determined by the spectra of the TFP conjugates. The number of dyes directly attached to the iron oxide nanoparticle was taken from the absorption at 755 nm using an extinction coefficient of 250000 M^–1^·cm^−1^ and assuming 2064 iron atoms per iron oxide nanoparticle. The number of dye-labeled peptides were taken from the absorption at 555 nm using an extinction coefficient of 104000 M^−^1·cm^−1^ for the M-peptide peptide substrate and from the absorption at 675 nm using an extinction coefficient of 190000 M^−^1·cm^−1^ for the C-peptide peptide substrate.

### 4.4. Fluorescence Activation of Peptide Substrates on TFP

#### 4.4.1. Fluorescence Activation with Pure Enzymes

The TFP probe was tested using purified CatB (Bovine, Calbiochem) and MMP2 (Human, Calbiochem) in a 96-well plate assay format. To test the fluorescence activation of TFP with CatB, 5–10 mg of the TFP probe was suspended in 150 mL of 20 mM sodium acetate buffer (pH 5), 2 mM DTT, and 5 mM EDTA with 10 mU of CatB. To test the fluorescence activation of TFP with MMP2, 5–10 mg of the TFP probe was suspended in 150 mL of 25 mM HEPES (pH 7) buffer containing 10 mM calcium chloride containing 0.5 mg of MMP2. The fluorescence activation of the TFP by CatB and MMP2 were measured in separate wells (i.e., the enzymes were not mixed in the same well) in a fluorescence microplate reader (Tecan, Safire II, Männedorf, Switzerland) at 37 ^o^C for 10 hrs. Fluorescence was monitored at 790 nm for the reporter fluorophore, 710 nm for the fluorescence activation of C-peptide (Cy5.5 dye), and 610 nm for the fluorescence activation of M-peptide (AF546 dye). Every 10 min, data points were acquired and then analyzed using Microsoft Excel.

#### 4.4.2. Fluorescence Activation in an in Vitro Whole Cell Assay

The TFP probe was tested in cell culture using the HT29 human cancer cell line. HT29 cells were cultured in McCoy’s 5a medium containing 10% fetal bovine serum (FBS) and 1% penicillin/streptomycin in a black 96-well plate with a clear glass coverslip bottom (Corning Costar). After 24 h, the cell culture medium was replaced with 200 mL of fresh medium containing 5–10 mg of the TFP probe and monitored in a fluorescence microplate reader (Safire II, Tecan) at 37 °C with data points taken every 10 min and then analyzed using Microsoft Excel.

### 4.5. Flow Cytometry Measurements

To test the cellular uptake of the TFP probe, human HT29 cancer cells were plated on 100 mm × 20 mm style tissue culture dishes in McCoy’s 5a medium with 10% FBS and 1% penicillin/streptomycin at 37°C. Once the cells were 75% confluent, the old media was aspirated, and new media containing ~0.5 mg TFP was added and incubated 18–24 h overnight at 37°C. The media was then aspirated and the cells were washed 3X with PBS. The cells were then removed via trypsin and washed again 3X and filtered prior to analysis using flow cytometry. The data were acquired on a BD Biosciences LSR II flow cytometer (BD Biosciences, San Jose, CA, USA) and analyzed with FlowJo software (Tree Star, Inc., Ashland, OR, USA).

### 4.6. Whole Animal Imaging Studies

The animal research protocol (Protocol #:2003N000209) has been reviewed and approved on August 20, 2003, by the Institutional Animal Care and Use Committee (IACUC–OLAW Assurance # D16-00361) of Massachusetts General Hospital. The protocol, as submitted and reviewed, conforms to the USDA Animal Welfare Act, Public Health Service (PHS) Policy on Humane Care and Use of Laboratory Animals, the “ILAR Guide for the Care and Use of Laboratory Animals” and other applicable laws and regulations. HT29 human colorectal adenocarcinoma cells (2 × 10^6^ cells per injection) were injected subcutaneously onto the flank of female nude (nu/nu) mice for the analysis of the TFP probe. Tissue signal intensities were determined after the intravenous (IV) injection of the TFP probe into the mice through the tail vein (2.0 mg Fe/kg). After 24 h, animals were imaged using a whole mouse imaging system (BonSAI, Seimens) with band pass filters for AF546 (ex/540 nm–560 nm, em/580 nm–600 nm), Cy5.5 (ex/620 nm–650 nm, em/680 nm–710 nm), and AF750 (ex/720 nm–760 nm, em/780 nm–820 nm) that monitored the fluorescence activation of the M-peptide substrate (AF546), C-peptide substrate (Cy5.5), and reporter dye, respectively. All filters were obtained from Omega Optical (Brattleboro, VT, USA). Images were analyzed using Osirix imaging software. Animals were handled according to institutional guidelines.

## Figures and Tables

**Figure 1 ijms-21-03068-f001:**
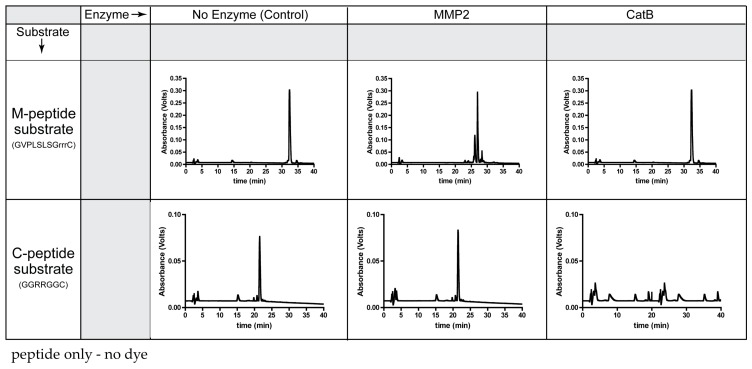
Specificity of each peptide substrate for its corresponding enzyme.

**Figure 2 ijms-21-03068-f002:**
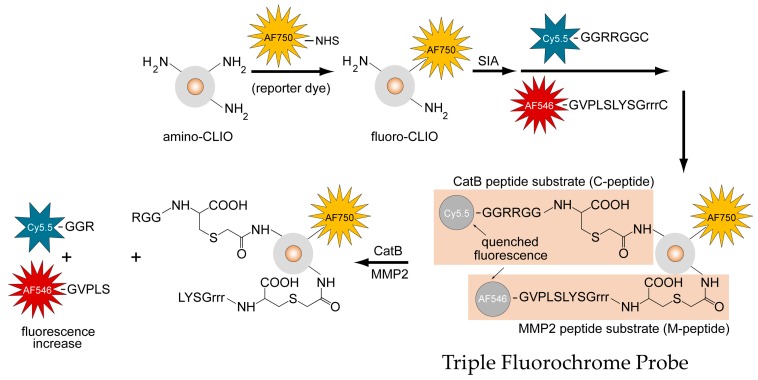
Design, synthesis, and activation mechanism of a triple fluorochrome probe for MMP2 and CatB activities.

**Figure 3 ijms-21-03068-f003:**
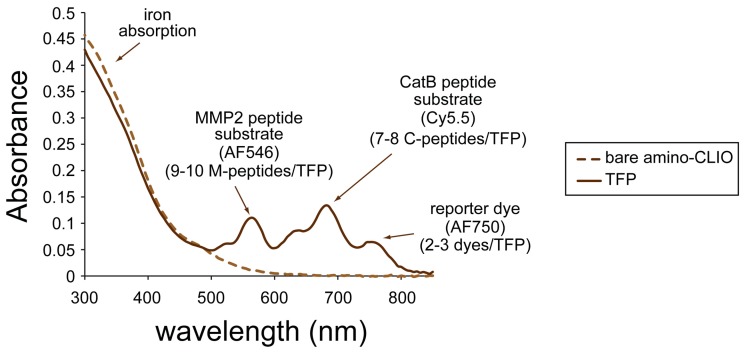
Ultraviolet-visible (UV-VIS) absorption spectrum of the triple fluorochrome probe.

**Figure 4 ijms-21-03068-f004:**
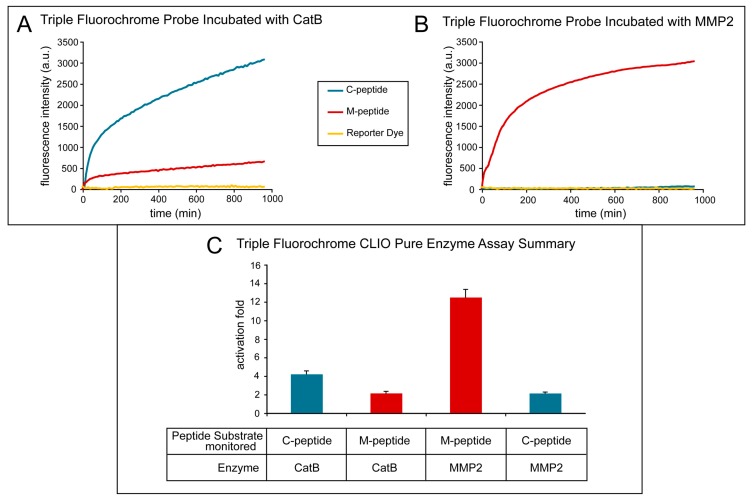
Activation of the triple fluorochrome probe by MMP2 and CatB enzymes (in vitro assay). (**A**) Intensities when incubated with CatB; (**B**) intensities when incubated with MMP2; (**C**) summary figure of signal intensity per peptide per incubation enzyme.

**Figure 5 ijms-21-03068-f005:**
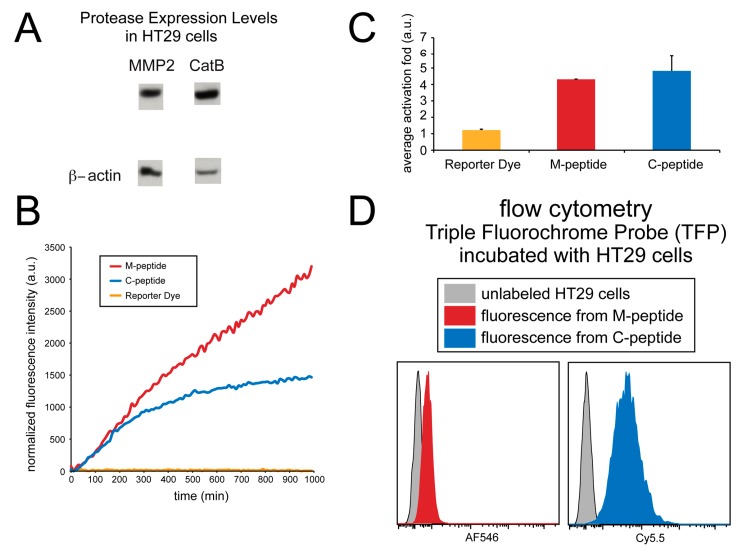
Detection of MMP2 and CatB activities in whole cells (HT29). (**A**) MMP2 and CatB expression immunoblot; β-actin was used as a control. (**B**) The fluorescence intensity of the probes over time. (**C**) The average fold-induction of fluorescence. (**D**) The flow cytometry of the TFP incubated with HT29 cells.

**Figure 6 ijms-21-03068-f006:**
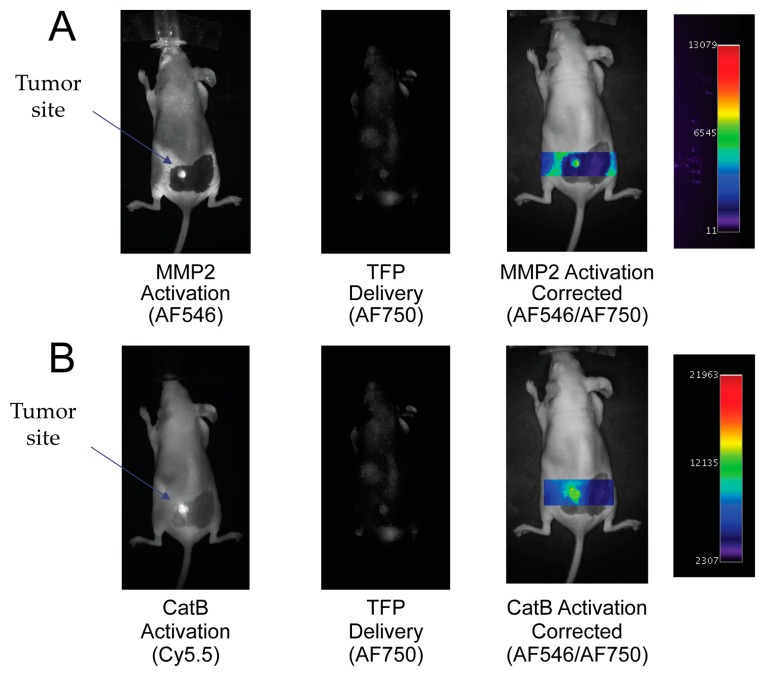
In vivo imaging of MMP2 and CatB activities with the Triple Fluorochrome Probe (TFP). (**A**) Imaging of MMP2 and the delivery reporter. (**B**) Imaging of CatB and the delivery reporter.

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
