# Peer review of "Simultaneous Monitoring of Multi-Enzyme Activity and Concentration in Tumor Using a Triply Labeled Fluorescent In Vivo Imaging Probe"

_ijms, 2020, doi:10.3390/ijms21093068_

Round 1

Reviewer 1 Report

Simultaneous Monitoring of Multi-Enzyme Activity 2 and Concentration in Prostate Cancer Using a Triply-3 Labeled Fluorescent In Vivo Imaging Probe. By Tam et al.

This study describes a new triple fluorochrome that contains one fluorochrome to report on the local substrate concentration and two fluorophores to monitor the local activity of two enzymes, cathepsin B (CatB) and matrix metalloprotease-2 (MMP2). Unlike previous synthetic strategies, the technique outlined in this MS pre-labels the peptide substrates prior to conjugation of the nanoparticle scaffold.

The MS is well written and logical. However, there are concerns regarding the approach and results that need to be considered.

Major points:

1- The in vitro experiment in figure 4 was performed by adding the MMP2 and CatB enzymes to test the activation of the TFP by the enzymes. This was then followed by a cell culture experiment (Fig 5) with HT29 cells, which express the MMP2 and CatB enzymes. It is unclear why the authors did not use a cell model with high and low expression of each of the enzymes or a genetic (shRNA) or pharmacological inhibitor of the enzymes to prove the point. Please clarify or provide additional experiments.

2- The results in figure 5 imply that CatB is more abundant (>2-fold) than MMP2, however, the fluorescence intensity on Fig 5B is higher for the M-peptide. This seems contradictory to the results in fig 4 about specificity of the probes. Please clarify.

3- It is unclear why HT29 were placed in female mice.

Minor points

1- Please explain or identity with arrows the important aspects of Figure 6. Is the bright spot the mouse bladder? What is the expected distribution of the delivery fluorophore?

2- It would be great if authors could comment on how deep into the tissues the TFP can still provide information.

Author Response

Reviewer 1’s critiques and Responses

Simultaneous Monitoring of Multi-Enzyme Activity 2 and Concentration in Prostate Cancer Using a Triply-3 Labeled Fluorescent In Vivo Imaging Probe. By Tam et al.

This study describes a new triple fluorochrome that contains one fluorochrome to report on the local substrate concentration and two fluorophores to monitor the local activity of two enzymes, cathepsin B (CatB) and matrix metalloprotease-2 (MMP2). Unlike previous synthetic strategies, the technique outlined in this MS pre-labels the peptide substrates prior to conjugation of the nanoparticle scaffold.

The MS is well written and logical. However, there are concerns regarding the approach and results that need to be considered.”

Major points:

  • The in vitro experiment in figure 4 was performed by adding the MMP2 and CatB enzymes to test the activation of the TFP by the enzymes. This was then followed by a cell culture experiment (Fig 5) with HT29 cells, which express the MMP2 and CatB enzymes. It is unclear why the authors did not use a cell model with high and low expression of each of the enzymes or a genetic (shRNA) or pharmacological inhibitor of the enzymes to prove the point. Please clarify or provide additional experiments.

Response: We agree that testing the probe in cellular conditions that should cause minimal fluorescence would be useful. However, taking our cellular results in tandem with the other experiments was deemed sufficient for this pilot study of our probe design. The excellent experiments suggested by the reviewer will be in our future studies.

  • The results in figure 5 imply that CatB is more abundant (>2-fold) than MMP2, however, the fluorescence intensity on Fig 5B is higher for the M-peptide. This seems contradictory to the results in fig 4 about specificity of the probes. Please clarify.

Response:  As noted in the text, the average activation fold is calculated based on the ending and initial fluorescence for each probe. The initial fluorescence for each probe was not the same; though the final intensity for MMP2 was higher than that of CatB, the activation fold of CatB was greater, as its initial fluorescence intensity was smaller.

  • It is unclear why HT29 were placed in female mice.

Response: The purpose of injecting HT29 cells in the mice was to examine the efficacy of the TFP in the context of a full-body organism. As prostate cancer is only for male, the female mice were used to avoid confounding factor- sex.

Minor points

  • Please explain or identity with arrows the important aspects of Figure 6. Is the bright spot the mouse bladder? What is the expected distribution of the delivery fluorophore?

Response: We do not have an expected distribution for our probe, but the bright spot is the tumor/site of HT29 cell injection.

  • It would be great if authors could comment on how deep into the tissues the TFP can still provide information.

Response: Though we did not investigate this in the present study, we agree that this would be a good focus for future analysis of the probe.

Reviewer 2 Report

In the current work, authors have synthesized a triply labeled fluorescent iron oxide nanoparticle imaging probe, including an AF750 substrate concentration reporter along with probes for cathepsin B (CatB) sand MMP-2 protease activity. The activity of the probe was verified through incubated with the proteases, and tested in vitro using the HT29 human prostate cancer cell line, and in vivo using female nude mice injected with HT29 cells. The probe had the appropriate specificity to the activity of their respective proteases, and the reporter dye did not activate when incubated in the presence of only MMP2 and CatB.

The study herein detailed constitutes an improvement of a previous design developed by the same authors. By using this method, authors are able to, simultaneously in vivo,

image multiple enzyme activities and multiple molecular parameters. In addition, the incorporation of a reporter (AF750) in the nanoparticle allows, unlike other designs, to correct the signal intensities of the protease probes (MMP2 and CatB).

From the methodological point of view the work is very well performed and the impact of these results in the field of clinical application are very promising.

Author Response

Reviewer 2’s critiques and Responses

“In the current work, authors have synthesized a triply labeled fluorescent iron oxide nanoparticle imaging probe, including an AF750 substrate concentration reporter along with probes for cathepsin B (CatB) sand MMP-2 protease activity. The activity of the probe was verified through incubated with the proteases, and tested in vitro using the HT29 human prostate cancer cell line, and in vivo using female nude mice injected with HT29 cells. The probe had the appropriate specificity to the activity of their respective proteases, and the reporter dye did not activate when incubated in the presence of only MMP2 and CatB.

The study herein detailed constitutes an improvement of a previous design developed by the same authors. By using this method, authors are able to, simultaneously in vivo, image multiple enzyme activities and multiple molecular parameters. In addition, the incorporation of a reporter (AF750) in the nanoparticle allows, unlike other designs, to correct the signal intensities of the protease probes (MMP2 and CatB).

From the methodological point of view the work is very well performed and the impact of these results in the field of clinical application are very promising.”

Response: We thank this reviewer for very encouraging comments upon our manuscript.